# Relation Prediction as an Auxiliary Training Objective for Improving Multi-Relational Graph Representations

**Yihong Chen**                                                  YIHONG.CHEN@CS.UCL.AC.UK
*University College London, London, United Kingdom*
*Facebook AI Research, London, United Kingdom*

**Pasquale Minervini**                                          P.MINERVINI@CS.UCL.AC.UK
*University College London, London, United Kingdom*

**Sebastian Riedel**                                            S.RIEDEL@CS.UCL.AC.UK
*University College London, London, United Kingdom*
*Facebook AI Research, London, United Kingdom*

**Pontus Stenetorp**                                            P.STENETORP@CS.UCL.AC.UK
*University College London, London, United Kingdom*

## Abstract

Learning good representations on multi-relational graphs is essential to knowledge base completion (KBC). In this paper, we propose a new self-supervised training objective for multi-relational graph representation learning, via simply incorporating *relation prediction* into the commonly used 1vsAll objective. The new training objective contains not only terms for predicting the subject and object of a given triple, but also a term for predicting the relation type. We analyse how this new objective impacts multi-relational learning in KBC: experiments on a variety of datasets and models show that *relation prediction* can significantly improve entity ranking, the most widely used evaluation task for KBC, yielding a 6.1% increase in MRR and 9.9% increase in Hits@1 on FB15k-237 as well as a 3.1% increase in MRR and 3.4% in Hits@1 on Aristo-v4. Moreover, we observe that the proposed objective is especially effective on highly multi-relational datasets, i.e. datasets with a large number of predicates, and generates better representations when larger embedding sizes are used.

## 1. Introduction

Aiming at completing missing entries, Knowledge Base Completion (KBC), also known as Link Prediction, plays a crucial role in constructing large-scale knowledge graphs [Nickel et al., 2016, Ji et al., 2020, Li et al., 2020]. Over the past years, most of the research on KBC has been focusing on Knowledge Graph Embedding models, which learn representations for all entities and relations in a Knowledge Graph, and use them for scoring whether an edge exists or not [Nickel et al., 2016]. Numerous models and architectural innovations have been proposed in the literature, including but not limited to translation-based models [Bordes et al., 2013], latent factorisation models [Nickel et al., 2011, Trouillon et al., 2016, Balazevic et al., 2019], and neural network-based models [Dettmers et al., 2018, Schlichtkrull et al., 2018, Xu et al., 2020].

Other more recent research has been making complementary efforts on analysing the evaluation procedures for these KBC models. For instance, Sun et al. [2020] call for standardisation of evaluation protocols; Kadlec et al. [2017], Ruffinelli et al. [2020] and Jain et al. [2020] highlight the importance of training strategies and show that careful hyper-parameter tuning can produce more accurate results

than adopting more elaborate model architectures; Lacroix et al. [2018] suggests that a simple model can produce state-of-the-art results when its training objective is properly selected.

Taking inspiration from these findings, this paper explores *relation prediction*: a simple auxiliary training objective that significantly improves multi-relational graph representation learning across several KBC models. Aside from training models to predict the subject and object entities for triples in a Knowledge Graph, we also train them to predict *relation types*, leading to a self-supervised training objective. Intuitively, this approach is akin to using a masked language model-like training objective [Devlin et al., 2019] instead of the commonly used auto-regressive training objective for KBC. In our experiments, we find that the new auxiliary training objective significantly improves downstream link prediction accuracy.

Empirical evaluations on various models and datasets support the effectiveness of our new training objective: the largest improvements were observed on ComplEx-N3 [Trouillon et al., 2016] and CP-N3 [Lacroix et al., 2018] with embedding sizes between 2K and 4K, providing up to $9.9\%$ boost in Hits@1 and $6.1\%$ boost in MRR on FB15k-237 with negligible computational overhead.

We further experiment on datasets with varying numbers of predicates and find that relation prediction helps more when the dataset is highly multi-relational, i.e. contains a larger number of predicates. Moreover, our qualitative analysis demonstrates improved prediction of some MANY-TO-MANY [Bordes et al., 2013] predicates and more diversified relation representations.

## 2. Background and Related Work

A Knowledge Graph $\mathcal{G} \subseteq \mathcal{E} \times \mathcal{R} \times \mathcal{E}$ contains a set of subject-predicate-object $\langle s, p, o \rangle$ triples, where each triple represents a relationship of type $p \in \mathcal{R}$ between the subject $s \in \mathcal{E}$ and the object $o \in \mathcal{E}$ of the triple. Here, $\mathcal{E}$ and $\mathcal{R}$ denote the set of all entities and relation types, respectively.

**Knowledge Graph Embedding Models**   A Knowledge Graph Embedding (KGE) model, also referred to as *neural link predictor*, is a differentiable model where entities in $\mathcal{E}$ and relation types in $\mathcal{R}$ are represented in a continuous embedding space, and the likelihood of a link between two entities is a function of their representations. More formally, KGE models are defined by a parametric *scoring function* $\phi_\theta : \mathcal{E} \times \mathcal{R} \times \mathcal{E} \mapsto \mathbb{R}$, with parameters $\theta$ that, given a triple $\langle s, p, o \rangle$, produces the likelihood that entities $s$ and $o$ are related by the relationship $p$.

**Scoring Functions**   KGE models can be characterised by their scoring function $\phi_\theta$. For example, in TransE [Bordes et al., 2013], the score of a triple $\langle s, p, o \rangle$ is given by $\phi_\theta(s, p, o) = - \left\| \mathbf{s} + \mathbf{p} - \mathbf{o} \right\|_2$, where $\mathbf{s}, \mathbf{p}, \mathbf{o} \in \mathbb{R}^k$ denote the embedding representations of $s$, $p$, and $o$, respectively. In Dist-Mult [Yang et al., 2015], the scoring function is defined as $\phi_\theta(s, p, o) = \langle \mathbf{s}, \mathbf{p}, \mathbf{o} \rangle = \sum_{i=1}^k \mathbf{s}_i \mathbf{p}_i \mathbf{o}_i$, where $\langle \cdot, \cdot, \cdot \rangle$ denotes the tri-linear dot product. Canonical Tensor Decomposition [CP, Hitchcock, 1927] is similar to DistMult, with the difference that each entity $x$ has two representations, $\mathbf{x}_s \in \mathbb{R}^k$ and $\mathbf{x}_o \in \mathbb{R}^k$, depending on whether it is being used as a subject or object: $\phi_\theta(s, p, o) = \langle \mathbf{s}_s, \mathbf{p}, \mathbf{o}_o \rangle$. In RESCAL [Nickel et al., 2011], the scoring function is a bilinear model given by $\phi_\theta(s, p, o) = \mathbf{s}^\top \mathbf{P} \mathbf{o}$, where $\mathbf{s}, \mathbf{o} \in \mathbb{R}^k$ is the embedding representation of $s$ and $p$, and $\mathbf{P} \in \mathbb{R}^{k \times k}$ is the representation of $p$. Note that DistMult is equivalent to RESCAL if $\mathbf{P}$ is constrained to be diagonal. Another variation of this model is ComplEx [Trouillon et al., 2016], where the embedding representations of $s$, $p$, and $o$ are complex vectors – i.e. $\mathbf{s}, \mathbf{p}, \mathbf{o} \in \mathbb{C}^k$ – and the scoring function is given by $\phi_\theta(s, p, o) = \Re(\langle \mathbf{s}, \mathbf{p}, \overline{\mathbf{o}} \rangle)$, where $\Re(\mathbf{x})$ represents the real part of $\mathbf{x}$, and $\overline{\mathbf{x}}$ denotes the complex conjugate of $\mathbf{x}$. In TuckER [Balazevic et al., 2019], the scoring function is

defined as $\phi_\theta(s, p, o) = \mathbf{W} \times_1 \mathbf{s} \times_2 \mathbf{p} \times_3 \mathbf{o}$, where $\mathbf{W} \in \mathbb{R}^{k_s \times k_p \times k_o}$ is a three-way tensor of parameters, and $\mathbf{s} \in \mathbb{R}^{k_s}$, $\mathbf{p} \in \mathbb{R}^{k_p}$, and $\mathbf{o} \in \mathbb{R}^{k_o}$ are the embedding representations of $s$, $p$, and $o$. In this work, we mainly focus on DistMult, CP, ComplEx, and TuckER, due to their effectiveness on several link prediction benchmarks [Ruffinelli et al., 2020, Jain et al., 2020].

**Training Objectives**    Another dimension for characterising KGE models is their *training objective*. Early tensor factorisation models such as RESCAL and CP were trained to minimise the reconstruction error of the whole adjacency tensor [Nickel et al., 2011]. To scale to larger Knowledge Graphs, subsequent approaches such as Bordes et al. [2013] and Yang et al. [2015] simplified the training objective by using *negative sampling*: for each training triple, a corruption process generates a batch of negative examples by corrupting the subject and object of the triple, and the model is trained by increasing the score of the training triple while decreasing the score of its corruptions. This approach was later extended by Dettmers et al. [2018] where, given a subject $s$ and a predicate $p$, the task of predicting the correct objects is cast as a $|\mathcal{E}|$-dimensional multi-label classification task, where each label corresponds to a distinct object and multiple labels can be assigned to the $(s, p)$ pair. This approach is referred to as KvsAll by Ruffinelli et al. [2020]. Another extension was proposed by Lacroix et al. [2018] where, given a subject $s$ and an object $p$, the task of predicting the correct object $o$ in the training triple is cast as a $|\mathcal{E}|$-dimensional multi-class classification task, where each class corresponds to a distinct object and only one class can be assigned to the $(s, p)$ pair. This is referred to as 1vsAll by Ruffinelli et al. [2020].

Note that, for factorisation-based models like DistMult, ComplEx, and CP, KvsAll and 1vsAll objectives can be computed efficiently on GPUs [Lacroix et al., 2018, Jain et al., 2020]. For example for DistMult, the score of all triples with subject $s$ and predicate $p$ can be computed via $\mathbf{E}(\mathbf{s} \odot \mathbf{p})$, where $\odot$ denotes the element-wise product, and $\mathbf{E} \in \mathbb{R}^{|\mathcal{E}| \times k}$ is the entity embedding matrix. In this work, we follow Lacroix et al. [2018] and adopt the 1vsAll loss, so as to be able to compare with their results, and since Ruffinelli et al. [2020] showed that they produce similar results in terms of downstream link prediction accuracy.

Recent work on standardised evaluation protocols for KBC models [Sun et al., 2020] and their systematic evaluation [Kadlec et al., 2017, Mohamed et al., 2019, Jain et al., 2020, Ruffinelli et al., 2020] shows that latent factorisation based models such as RESCAL, ComplEx, and CP are very competitive when their hyper-parameters are tuned properly [Kadlec et al., 2017, Ruffinelli et al., 2020]. In this work, we show that using relation prediction as an auxiliary training task can further improve their downstream link prediction accuracy.

## 3. Relation Prediction as An Auxiliary Training Objective

In what is referred to as the 1vsAll setting [Ruffinelli et al., 2020], KBC models are trained using a self-supervised training objective by maximising the conditional likelihood of the subject $s$ (resp. object $o$) of training triples, given the predicate and the object $o$ (resp. subject $s$). More formally,

KBC models are trained by maximising the following objective:

$$\arg\max_{\theta\in\Theta} \sum_{\langle s,p,o\rangle\in\mathcal{G}} \left[\log P_\theta(s\mid p,o) + \log P_\theta(o\mid s,p)\right]$$

$$\text{with}\quad \log P_\theta(o\mid s,p) = \phi_\theta(s,p,o) - \log\sum_{o'\in\mathcal{E}} \exp\left[\phi_\theta(s,p,o')\right]$$

$$\log P_\theta(s\mid p,o) = \phi_\theta(s,p,o) - \log\sum_{s'\in\mathcal{E}} \exp\left[\phi_\theta(s',p,o)\right], \tag{1}$$

where $\theta\in\Theta$ are the model parameters, including entity and relation embeddings, and $\phi_\theta$ is a scoring function parameterised by $\theta$. Such an objective limits predicting positions in the training objective to either the first ($s$) or the last ($o$) element of the triple.

In this work, we propose relation prediction as an auxiliary task for training KBC models. The new training objective not only contains terms for predicting the subject and the object of the triple – $\log P(s\mid p,o)$ and $\log P(o\mid s,p)$ in Equation (1) – but also an objective $\log P(p\mid s,o)$ for predicting the relation type $p$:

$$\arg\max_{\theta\in\Theta} \sum_{\langle s,p,o\rangle\in\mathcal{G}} \left[\log P_\theta(s\mid p,o) + \log P_\theta(o\mid s,p) + \lambda\log P_\theta(p\mid s,o)\right]$$

$$\text{with}\quad \log P_\theta(p\mid s,o) = \phi_\theta(s,p,o) - \log\sum_{p'\in\mathcal{R}} \exp\left[\phi_\theta(s,p',o)\right], \tag{2}$$

where $\lambda\in\mathbb{R}_+$ is a user-specified hyper-parameter that determines the contribution of the relation prediction objective; we assume $\lambda=1$ unless specified otherwise. This new training objective adds very little overhead to the training process, and can be easily added to existing KBC implementations; PyTorch examples are included in Appendix A. Compared to conventional approaches, relation prediction can help the model learn to further distinguish among different predicates.

## 4. Empirical Study

In this section, we conduct several experiments to verify the effectiveness of incorporating relation prediction as an auxiliary training objective. We are interested in the following research questions:

**RQ1:** How does the new training objective impact the results of downstream knowledge base completion tasks across different datasets? How does the number of relation types on the datasets affect the new training objective?

**RQ2:** How does the new training objective impact different models? Does it benefit all the models uniformly, or it particularly helps some of them?

**RQ3:** Does the new training objective produce better entity and relation representations?

**Datasets** We use Nations, UMLS, and Kinship from Kok and Domingos [2007], WN18RR [Dettmers et al., 2018], and FB15k-237 [Toutanova et al., 2015], which are commonly used in the KBC literature. As these datasets contain a relatively small number of predicates, we also experiment with Aristo-v4, the 4-th version of Aristo Tuple KB [Mishra et al., 2017], which has more than 1.6k predicates. Since Aristo-v4 has no standardised splits for KBC, we randomly sample 20k triples for test and 20k for validation. Table 1 summarises the statistics of these datasets.

| Dataset | $|\mathcal{E}|$ | $|\mathcal{R}|$ | #Train | #Validation | #Test |
|---|---|---|---|---|---|
| Nations | 14 | 55 | 1,592 | 100 | 301 |
| UMLS | 135 | 46 | 5,216 | 652 | 661 |
| Kinship | 104 | 25 | 8,544 | 1,068 | 1,074 |
| WN18RR | 40,943 | 11 | 86,835 | 3,034 | 3,134 |
| FB15k-237 | 27,395 | 237 | 272,115 | 17,535 | 20,466 |
| Aristo-v4 | 44,950 | 1,605 | 242,594 | 20,000 | 20,000 |

Table 1: Dataset statistics, where $|\mathcal{E}|$ and $|\mathcal{R}|$ indicate the numbers of entities and predicates.

**Metrics**   Entity ranking is the most commonly used evaluation protocol for knowledge base completion. For a given query $(s, p, ?)$ or $(?, p, o)$, all the candidate entities are ranked based on the scores produced by the models, and the resulting ordering is used to compute the *rank* of the true answer. We use the standard filtered Mean Reciprocal Rank (MRR) and Hits@$K$ (Hit ratios of the top-K ranked results), with $K \in \{1, 3, 10\}$, as metrics.

**Models**   We use several competitive and reproducible [Ruffinelli et al., 2020, Sun et al., 2020] models: RESCAL [Nickel et al., 2011], ComplEx [Trouillon et al., 2016], CP [Lacroix et al., 2018], and TuckER [Balazevic et al., 2019]. To ensure fairness in various comparisons, we did an extensive tuning of hyper-parameters using the validation sets, which consists of 41,316 training runs in total. For the main results on all the datasets, we tuned $\lambda$ using grid-search. For the ablation experiments on the number of predicates and for different choices of models, we set $\lambda$ to 1 for simplicity. Details regarding the hyper-parameter sweeps can be found in Appendix B.

## 4.1 RQ1: Impacts of Relation Prediction on Different Datasets

How does the proposed training objective impact knowledge base completion on different datasets? To answer this research question, we compare the performance of training with relation prediction and training without relation prediction on several popular KBC datasets. For the smaller datasets (Kinship, Nations and UMLS), we selected the best one from RESCAL, ComplEx, CP, and TuckER. For larger datasets (WN18RR, FB15k-237, and Aristo-v4), due to a limited computation budget, we used ComplEx, which outperformed other models in our preliminary experiments.

Table 2 summarises the results on the smaller datasets, where ✔ indicates training with relation prediction while ✗ indicates training without relation prediction. We can observe that relation prediction brings a 2% – 4% improvement in MRR and Hits@1, as well as keeping a competitive Hits@3 and Hits@10.

Table 3 summarises the results on the larger datasets. Including relation prediction as an auxiliary training objective brings a consistent improvement on the 3 datasets with respect to all metrics, except for Hits@10 on WN18RR. Particularly, relation prediction leads to increases of $6.1\%$ in MRR, $9.9\%$ in Hits@1, $6.1\%$ in Hits@3 on FB15k-237 and $3.1\%$ in MRR, $3.4\%$ in Hits@1, $3.8\%$ in Hits@3 on Aristo-v4. Compared to WN18RR, we observe a larger improvement on FB15k-237 and Aristo-v4. One potential reason is that on FB15k-237 there is a more diverse set of predicates ($|\mathcal{R}| = 237$) and Aristo-v4 ($|\mathcal{R}| = 1605$) than in WN18RR ($|\mathcal{R}| = 11$). The number of predicates $|\mathcal{R}|$ on WN18RR is comparatively small, and the model gains more from distinguishing different entities than distinguishing relations. In other words, using lower values for $\lambda$ (the weight of the

| Dataset | Entity Prediction | Relation Prediction | MRR | Hits@1 | Hits@3 | Hits@10 |
|---|---|---|---|---|---|---|
| Kinship | ✗ | ✔ | **0.920** | **0.867** | **0.970** | **0.990** |
|  | ✔ | ✗ | 0.897 | 0.835 | 0.955 | 0.987 |
|  | ✔ | ✔ | 0.916 | 0.866 | 0.964 | 0.988 |
| Nations | ✗ | ✔ | 0.686 | 0.493 | 0.871 | 0.998 |
|  | ✔ | ✗ | 0.813 | 0.701 | **0.915** | **1.000** |
|  | ✔ | ✔ | **0.827** | **0.726** | **0.915** | 0.998 |
| UMLS | ✗ | ✔ | 0.863 | 0.795 | 0.914 | 0.979 |
|  | ✔ | ✗ | 0.960 | 0.930 | **0.991** | **0.998** |
|  | ✔ | ✔ | **0.971** | **0.954** | 0.986 | 0.997 |

Table 2: Test performance comparison on Kinship, Nations, and UMLS. We conducted an extensive hyper-parameter search over 4 different models, namely RESCAL, ComplEx, CP, and TuckER, where the model is also treated as a hyper-parameter. Including relation prediction as an auxiliary training objective on these three datasets helps in terms of test MRR and Hits@1, while keeping competitive test Hits@3 and Hits@10. More details on the hyper-parameter selection process are available in Appendix B.1.1.

| Dataset | Entity Prediction | Relation Prediction | MRR | Hits@1 | Hits@3 | Hits@10 |
|---|---|---|---|---|---|---|
| WN18RR | ✗ | ✔ | 0.258 | 0.212 | 0.290 | 0.339 |
|  | ✔ | ✗ | 0.487 | 0.441 | 0.501 | **0.580** |
|  | ✔ | ✔ | **0.488** | **0.443** | **0.505** | 0.578 |
| FB15K-237 | ✗ | ✔ | 0.263 | 0.187 | 0.287 | 0.411 |
|  | ✔ | ✗ | 0.366 | 0.271 | 0.401 | 0.557 |
|  | ✔ | ✔ | **0.388** | **0.298** | **0.425** | **0.568** |
| Aristo-v4 | ✗ | ✔ | 0.169 | 0.120 | 0.177 | 0.267 |
|  | ✔ | ✗ | 0.301 | 0.232 | 0.324 | 0.438 |
|  | ✔ | ✔ | **0.311** | **0.240** | **0.336** | **0.447** |

Table 3: Test performance comparison on WN18RR, FB15k-237, and Aristo-v4 using ComplEx. Including relation prediction as an auxiliary training objective brings consistent improvements across the three datasets on all metrics except Hits@10 on WN18RR. On FB15k-237 and Aristo-v4, adding relation prediction yields larger improvements in downstream link prediction tasks. More details on the hyper-parameter selection process are available in Appendix B.1.2.

relation prediction objective) is more suitable for datasets with fewer predicates but a large number of entities. We include ablations on $|\mathcal{R}|$ in Section 4.1.2.

| Dataset | MRR | Hits@1 | Hits@3 | Hits@10 |
|---|---|---|---|---|
| WN18RR | (15.0, 0.03125) | (15.0, 0.03125) | (15.0, 0.03125) | (3.0, 0.76740) |
| FB15k-237 | (15.0, 0.03125) | (15.0, 0.03125) | (15.0, 0.03125) | (15.0, 0.03125) |
| Aristo-v4 | (15.0, 0.03125) | (15.0, 0.03125) | (15.0, 0.03125) | (15.0, 0.03125) |

Table 4: Wilcoxon signed-rank test for ComplEx-N3 on several datasets. For each dataset and metric, we report the corresponding statistics – i.e. the sum of ranks of positive differences – and the p-value as (statistics, p-value).

### 4.1.1 SIGNIFICANCE TESTING

In order to show that the improvements brought by relation perturbation are significant, we run the experiments with 5 random seeds and perform Wilcoxon signed-rank test [Wilcoxon, 1992] over the metrics obtained with and without relation prediction. The test is performed as follows. First, we computed the differences between results obtained with ComplEx trained with and without relation prediction. The null hypothesis is that the median of the differences is negative. Table 4 summarises the result. We can observe that almost all p-values are about 0.03, which means we can reject the null hypothesis at a confidence level of about 97%. The new training objective that incorporates relation prediction as an auxiliary training objective significantly improves the performance of KBC models except for Hits@10 on WN18RR.

### 4.1.2 ABLATION ON THE NUMBER OF PREDICATES

As previously discussed, relation prediction brings different impacts to WN18RR, FB15k-237, and Aristo-v4. Since one of the biggest differences among these datasets is the number of different predicates $|\mathcal{R}|$ ($1,605$ for Aristo-v4 and 237 for FB15k-237, while only 11 for WN18RR), we would like to determine the impact of perturbing relations with various $|\mathcal{R}|$. In order to achieve this, we construct a series of datasets with different $|\mathcal{R}|$ by sampling triples containing a subset of predicates from FB15k-237. For example, to construct a dataset with only 5 predicates, we first sampled 5 predicates from the set of 237 predicates and then extracted triples containing these 5 predicates as the new dataset. In total, we have datasets with $|\mathcal{R}| \in [5, 25, 50, 100, 150, 200]$ predicates. To address the noise introduced in predicate sampling during datasets construction, we experimented with 3 random seeds. For convenience, we set the weight of relation prediction $\lambda$ to 1 and performed a similar grid-search over the regularisation and other hyper-parameters to ensure that the models were regularised and trained appropriately with the different amounts of training and test data points.

Results are summarised in Figure 1. As shown in the right portion of Figure 1, predicting relations helps datasets with more predicates, resulting in a 2%–4% boost in MRR, Hits@1, and Hits@3. For datasets with fewer than 50 predicates, there is considerable fluctuation in the relative change as shown in the left portion of the figure – but a clear downward trend. These results verify our hypothesis that relation prediction brings benefits to datasets with a larger number of predicates. Note that we did not tune the weight of relation prediction objective $\lambda$ (and fixed it to 1), and this choice might have been sub-optimal on datasets with a fewer number of predicates.

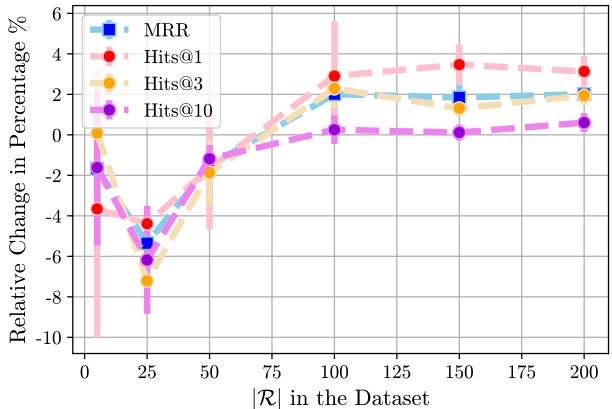

Figure 1: Relative changes between ComplEx trained with and w/o Relation Prediction on datasets with varying numbers of predicates $|\mathcal{R}|$. We experimented with 3 random seeds. Larger bars mean more variance. Relative changes were computed with $(m_+ - m_-)/m_-$, where $m_+$ and $m_-$ denote the metric values with and w/o relation prediction. A clear downward trend can be observed for datasets with $|\mathcal{R}| < 50$ while $2\% - 4\%$ clear increase in MRR, Hits@1, and Hits@3 are shown where $|\mathcal{R}| > 50$.

| Model | Relation Prediction | MRR | Hits@1 | Hits@3 | Hits@10 |
|---|---|---|---|---|---|
| CP | ✗ | 0.356 | 0.262 | 0.392 | 0.546 |
|  | ✔ | **0.366** | **0.274** | **0.401** | **0.550** |
| ComplEx | ✗ | 0.366 | 0.271 | 0.401 | 0.557 |
|  | ✔ | **0.382** | **0.289** | **0.419** | **0.568** |
| RESCAL | ✗ | 0.356 | 0.266 | 0.390 | 0.532 |
|  | ✔ | **0.359** | **0.271** | **0.395** | **0.533** |
| TuckER | ✗ | 0.351 | 0.260 | 0.386 | 0.532 |
|  | ✔ | **0.354** | **0.264** | **0.388** | **0.535** |

Table 5: Test performance comparison on FB15k-237 across 4 different models – CP, ComplEx, RESCAL, and TuckER. We set the weight of relation prediction to 1 and performed a grid search over hyper-parameters. More details are available in the appendix. While relation prediction seems to help all 4 models, it brings more benefit to CP and ComplEx compared to TuckER and RESCAL.

## 4.2 RQ2: Impacts of Relation Prediction on Different KBC Models

For measuring how does relation prediction influences the downstream accuracy of KBC models, we run experiments on FB15k-237 with several models – namely ComplEx, CP, TuckER, and RESCAL.

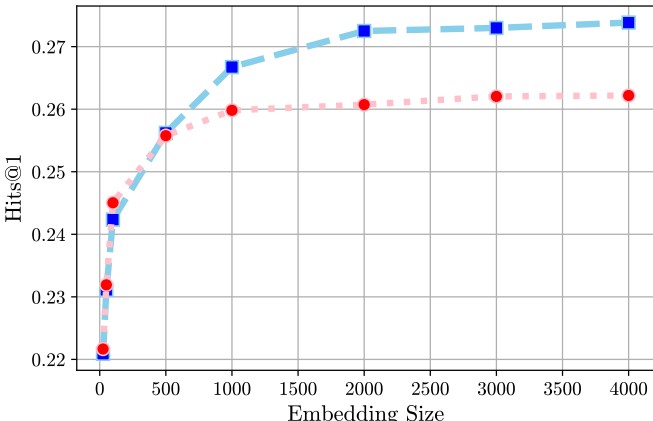

Figure 2: Hits@1 versus embedding size for CP on FB15k-237, each point represents a model trained with some specific embedding size with (blue) / -out (red) perturbing relations. The smallest embedding size is 25.

For simplicity, we set the weight of relation prediction $\lambda$ to 1. As shown in Table 5, including relation prediction as an auxiliary training objective brings consistent improvement to all models. Notably, up to a 4.4% and a 6.6% increase in Hits@1 can be observed respectively for CP and ComplEx. For TuckER and RESCAL, the improvements brought by relation perturbation are relatively small. This may be due to the fact that we had to use smaller embedding sizes for TuckER and RESCAL, since these models are known to suffer from scalability problems when used with larger embedding sizes. We include the ablation on embedding sizes of the models in Section 4.2.1. As for the computational cost, in our experiments, adopting the new loss only added an average 2% increase in training time per epoch, though it might require more epochs to converge.

### 4.2.1 ABLATIONS ON EMBEDDING SIZES

In our experiments, increasing the embedding size of the model leads to better performance. However, there might exist a saturating point where larger embedding sizes stop boosting the performance. We are interested in how perturbing relations will impact the saturating point and which embedding sizes benefit most from it. Figure 2 shows the relationship between the embedding size and the MRR for CP on FB15k-237. At small embedding sizes, perturbing relations makes little difference. However, it does help CP with larger embedding sizes and delays the saturating point. As we can see, the slope of the blue curve is larger than the red one, which bends little between an embedding size of 1,000 and an embedding size of 4,000. We can thus observe that perturbing relations leaves more headroom to improve the model by increasing embedding sizes.

### 4.3  RQ3: Qualitative Analysis of the Learned Entity and Relation Representations

In our experiments, we observe that relation prediction improves the link prediction accuracy for MANY-TO-MANY predicates, which are known to be difficult for KBC models [Bordes et al., 2013]. Table 6 lists the top 10 predicates that benefit most from relation prediction. We rank the predicates

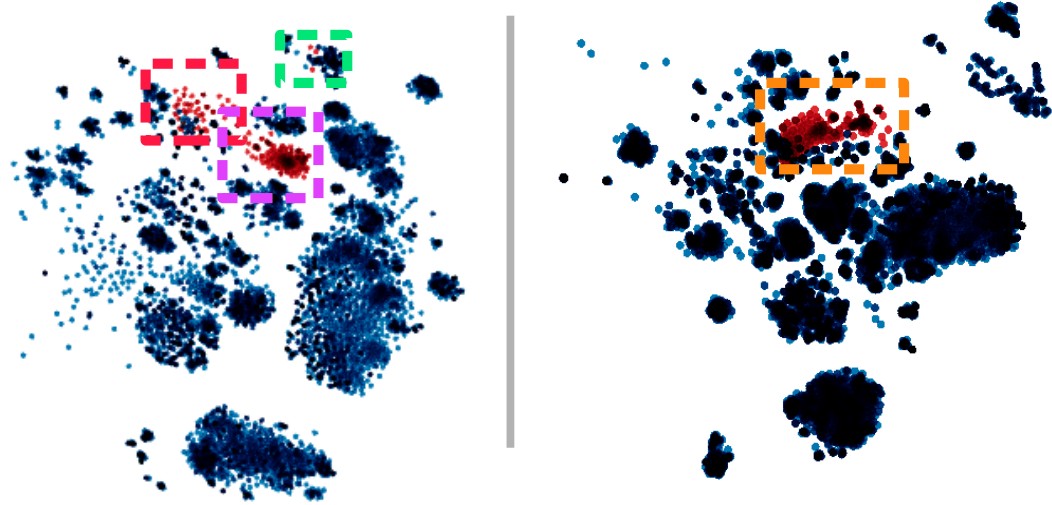

Figure 3: t-SNE visualisations for ComplEx embeddings, trained with relation prediction (left) and without relation prediction (right). Red points and blue points correspond to predicates and entities respectively. Dashed boxes highlight different clusters.

| |
|---|
| /ice_hockey/hockey_team/current_roster./sports/sports_team_roster/position |
| /sports/sports_team/roster./baseball/baseball_roster_position/position |
| /location/country/second_level_divisions |
| /tv/tv_producer/programs_produced./tv/tv_producer_term/program |
| /olympics/olympic_sport/athletes./olympics/olympic_athlete_affiliation/olympics |
| /award/award_winning_work/awards_won./award/award_honor/honored_for |
| /music/instrument/family |
| /olympics/olympic_games/sports |
| /base/biblioness/bibs_location/state |
| /soccer/football_team/current_roster./soccer/football_roster_position/position |

Table 6: Top 10 predicates that are improved most by relation prediction.

by averaging the associated MRR of $(s, p, ?)$ and $(?, p, o)$ queries. Table 14 and Table 15 list the top 20 queries of $(s, p, ?$ and $(?, p, o)$ that are improved most by relation prediction. We can see that relation prediction helps the queries like *"Where was film Magic Mike released?"*, *"Where was Paramount Pictures founded?"*, *"Which person appear in the film The Dictator 2012?"*, *"Which places are located in UK?"* and *"Which award did Vera Drake win?"*.

To intuitively understand why it helps with these predicates, we ran t-SNE over the learned entity and predicate representations. Reciprocal predicates are also included in the t-SNE visualisations. We set the embedding size to 1000, and use N3 regularisation. Hyper-parameters were chosen based on the validation MRR. We run t-SNE for 5000 steps with 50 as perplexity. As we can see from Figure 3, there are more predicate clusters in the t-SNE visualisation for relation prediction compared

to without relation prediction. This demonstrates relation prediction helps the model distinguish between different predicates: Most predicates are separated from the entities (the pink region) while some predicates with similar semantics or subject-object contexts form a cluster (the red region); There are also a few predicates, which are not close to their predicate counterparts but instead close to highly related entities (the green region). Table 7 lists 3 example predicates for each region. Though there can be information loss during the process of projecting high-dimensional embedding vectors into 2-dimensional space, we hope this visualisation will help illustrate how relation prediction helps to learn more diversified predicate representations.

| Pink Region |
| --- |
| /base/schemastaging/organization_extra/phone_number./base/schemastaging/phone_sandbox/contact_category |
| /location/statistical_region/places_exported_to./location/imports_and_exports/exported_to |
| /sports/sports_league/teams./sports/sports_league_participation/team |

| Red Region |
| --- |
| /people/person/nationality |
| /people/person/religion |
| /soccer/football_team/current_roster./sports/sports_team_roster/position |

| Green Region |
| --- |
| /education/educational_institution/students_graduates./education/education/student |
| /common/topic/webpage./common/webpage/category |
| /education/educational_institution/students_graduates./education/education/major_field_of_study |

Table 7: Three example predicates in each region of the t-SNE plot.

## 5. Discussion and Conclusion

In this paper, we propose to use a new self-supervised objective for training KBC models - by simply incorporating *relation prediction* into the commonly used 1vsAll objective. In our experiments, we show that adding such a simple learning objective is significantly helpful to various KBC models. It brings up to $9.9\%$ boost in Hits@1 for ComplEx trained on FB15k-237, even though the evaluation task of entity ranking might seem irrelevant to *relation prediction*.

Our work suggests a worthwhile direction towards devising relation-aware self-supervised objectives for KBC. In this paper, we mainly focus on simple factorisation-based models. Future work will consider analysing the proposed objective for more complex KBC models, such as graph neural network-based KBC models, and on more datasets. Another interesting future work direction is analysing the proposed auxiliary objective on more downstream applications besides link prediction, and evaluate whether it can be used to learn useful multi-relational graph representations.

## Acknowledgements

We would like to thank the reviewers for their constructive feedback. We would also want to thank all the other members in UCL NLP for their support throughout COVID. PM and PS are supported by the European Union's Horizon 2020 research and innovation programme under grant agreement no. 875160.

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

## Appendix A. Code Snippet for Relation Prediction as an Auxiliary Training Objective

Figure 4 demonstrates how to add relation prediction to the existing implementation of ComplEx.

## Appendix B. Hyper-parameters Sweeps

In this section, we summarise all the hyper-parameters used in our experiments. We used Tesla P100 and Tesla V100 GPUs to run the experiments. We implemented each model by PyTorch. Our codebase is based on https://github.com/facebookresearch/kbc.

### B.1  Hyper-parameter Ranges of Relation Prediction Across Datasets

B.1.1  KINSHIP, NATIONS, AND UMLS

| Model | $d$ or $(d, d_r)$ | $lr$ | $bsz$ | $reg$ |
|---|---|---|---|---|
| RESCAL | [50, 100, 200] | [0.1, 0.01] | [10, 50, 100, 500] | [0, 0.005, 0.01, 0.05, 0.1, 0.5] |
| ComplEx | [100, 200, 500, 1000] | [0.1, 0.01] | [10, 50, 100, 500] | [0, 0.005, 0.01, 0.05, 0.1, 0.5] |
| CP | [200, 400, 1000, 2000] | [0.1, 0.01] | [10, 50, 100, 500] | [0, 0.005, 0.01, 0.05, 0.1, 0.5] |
| TuckER | [(100, 25), (200, 25), (100, 50), (200, 50), (100, 100), (200, 100)] | [0.1, 0.01] | [10, 50, 100, 500] | [0, 0.005, 0.01, 0.05, 0.1, 0.5] |

Table 8:  Hyper-parameter Search Different KBC Models on Small Datasets (Kinship, Nations, UMLS). $d$ stands for embedding size. $d_r$ stands for a separate embedding size of relations. $lr$ is the learning rate. $bsz$ is the batch size. $reg$ is the regularization strength.

| Dataset | Relation Prediction | Entity Prediction | Model | $d$ | $d_r$ | $lr$ | $bsz$ | $reg$ | $\lambda$ | Dev MRR |
|---|---|---|---|---|---|---|---|---|---|---|
| KINSHIP | ✔ | ✘ | Tucker | 200 | 100 | 0.10 | 10 | 0.1 | NA | 0.919581 |
|  | ✘ | ✔ | CP | 2000 | NA | 0.10 | 50 | 0.01 | NA | 0.897429 |
|  | ✔ | ✔ | CP | 2000 | NA | 0.10 | 50 | 0.05 | 4.000 | 0.918323 |
| NATIONS | ✔ | ✘ | Tucker | 200 | 50 | 0.01 | 10 | 0.1 | NA | 0.686010 |
|  | ✘ | ✔ | CP | 2000 | NA | 0.01 | 10 | 0.01 | NA | 0.855388 |
|  | ✔ | ✔ | TuckER | 200 | 25 | 0.01 | 10 | 0.10 | 0.250 | 0.865352 |
| UMLS | ✔ | ✘ | CP | 1000 | NA | 0.1 | 500 | 0.01 | NA | 0.863008 |
|  | ✘ | ✔ | ComplEx | 1000 | NA | 0.10 | 10 | 0.00 | NA | 0.967626 |
|  | ✔ | ✔ | ComplEx | 1000 | NA | 0.01 | 10 | 0.00 | 0.500 | 0.971612 |

Table 9:  Best Hyper-parameter Configuration and the Corresponding Validation MRR on Small Datasets. $d$ stands for embedding size. $d_r$ stands for a separate embedding size of relations. $lr$ is the learning rate. $bsz$ is the batch size. $reg$ is the regularization strength. $\lambda$ is the weighting of relation prediction. NA indicates not applicable.

For all small datasets (Kinship, Nations, UMLS), we trained RESCAL, ComplEx, CP and TuckER with Adagrad optimiser and N3 regularisation for at most 400 epochs. Reciprocal triples were included since they are reported to be helpful [Dettmers et al., 2018, Lacroix et al., 2018]. We did grid searches over hyper-parameter combinations and chose the best configuration for each

```python
1   class ComplEx(KBCModel):
2       def __init__(self, sizes, rank, init_size):
3           super(ComplEx, self).__init__()
4           self.sizes = sizes
5           self.rank = rank
6
7           self.embeddings = nn.ModuleList([
8               nn.Embedding(s, 2 * rank, sparse=False)
9               for s in sizes[:2]
10          ])
11          self.embeddings[0].weight.data *= init_size
12          self.embeddings[1].weight.data *= init_size
13
14      def forward(self, x, score_rhs=True, score_rel=False, score_lhs=False, normalize_rel=False):
15          lhs = self.embeddings[0](x[:, 0])
16          rel = self.embeddings[1](x[:, 1])
17          rhs = self.embeddings[0](x[:, 2])
18
19          lhs = lhs[:, :self.rank], lhs[:, self.rank:]
20          rel = rel[:, :self.rank], rel[:, self.rank:]
21          rhs = rhs[:, :self.rank], rhs[:, self.rank:]
22
23          rhs_scores, rel_scores = None, None
24          if score_rhs:
25              to_score_entity = self.embeddings[0].weight
26              to_score_entity = to_score_entity[:, :self.rank], to_score_entity[:, self.rank:]
27              rhs_scores = (
28                  (lhs[0] * rel[0] - lhs[1] * rel[1]) @ to_score_entity[0].transpose(0, 1) +
29                  (lhs[0] * rel[1] + lhs[1] * rel[0]) @ to_score_entity[1].transpose(0, 1)
30              )
31          if score_rel:
32              to_score_rel = self.embeddings[1].weight
33              to_score_rel = to_score_rel[:, :self.rank], to_score_rel[:, self.rank:]
34              rel_scores = (
35                  (lhs[0] * rhs[0] + lhs[1] * rhs[1]) @ to_score_rel[0].transpose(0, 1) +
36                  (lhs[0] * rhs[1] - lhs[1] * rhs[0]) @ to_score_rel[1].transpose(0, 1)
37              )
38          if score_lhs:
39              to_score_lhs = self.embeddings[0].weight
40              to_score_lhs = to_score_lhs[:, :self.rank], to_score_lhs[:, self.rank:]
41              lhs_scores = (
42                  (rel[0] * rhs[0] + rel[1] * rhs[1]) @ to_score_lhs[0].transpose(0, 1) +
43                  (rel[0] * rhs[1] - rel[1] * rhs[0]) @ to_score_lhs[1].transpose(0, 1)
44              )
```

Figure 4: Relation Prediction for ComplEx, the red region shows the lines related to using relation prediction as an auxiliary training task.

dataset based on validation MRR. We listed the grids of hyper-parameter search in Table 8 and report the best-searched configuration in Table 9. As for the balancing between relation prediction and entity prediction, we searched the weight of relation prediction over $\{4, 2, 0.5, 0.25, 0.125\}$.

### B.1.2 WN18RR, FB15K-237, AND ARISTO-V4

For all datasets, we trained ComplEx with N3 regularizer and Adagrad optimiser and N3 regularisation for at most 400 epochs. Reciprocal triples were included since they are reported to be helpful [Dettmers et al., 2018, Lacroix et al., 2018]. As for the weight of relation prediction, we searched over different zones on different datasets. For WN18RR, we searched the weight of relation prediction over $[0.005, 0.001, 0.05, 0.1, 0.5, 1]$. For FB15k-237, we searched over $[0.125, 0.25, 0.5, 1, 2, 4]$. For Aristo-v4, we searched over $[0.125, 0.25, 0.5, 1, 2, 4]$. We did grid searches over hyper-parameter combinations and chose the best configuration for each dataset based on validation MRR. We report the grids for each dataset in Table 10, and the best found configuration in Table 11.

| Dataset | $d$ | $lr$ | $bsz$ | $reg$ |
|---------|-----|------|-------|-------|
| WN18RR | [100, 500, 1000] | [0.1, 0.01] | [100, 500, 1000] | [0.005, 0.01, 0.05, 0.1, 0.5, 1] |
| FB15k-237 | [100, 500, 1000] | [0.1, 0.01] | [100, 500, 1000] | [0.0005, 0.005, 0.01, 0.05, 0.1, 0.5, 1, 0] |
| Aristo-v4 | [500, 1000, 1500] | [0.1, 0.01] | [100, 500, 1000] | [0, 0.005, 0.01, 0.05, 0.1, 0.5, 1] |

Table 10: Hyper-parameter Search for Vanila Relation Perturbation over ComplEx on Different Datasets $d$ stands for embedding size. $lr$ is the learning rate. $bsz$ is the batch size. $reg$ is the regularization strength. $\lambda$ is the weighting of relation prediction.

| Dataset | Relation Prediction | Entity Prediction | $d$ | $lr$ | $bsz$ | $reg$ | $\lambda$ | Dev MRR |
|---------|:---:|:---:|---|---|---|---|---|---|
| WN18RR | ✔ | ✗ | 1000 | 0.10 | 500 | 0.5 | NA | 0.257945 |
| | ✗ | ✔ | 1000 | 0.10 | 100 | 0.10 | NA | 0.488083 |
| | ✔ | ✔ | 1000 | 0.10 | 100 | 0.10 | 0.050 | 0.490053 |
| FB15k-237 | ✔ | ✗ | 1000 | 0.10 | 1000 | 0.0005 | NA | 0.262888 |
| | ✗ | ✔ | 1000 | 0.10 | 100 | 0.05 | NA | 0.372305 |
| | ✔ | ✔ | 1000 | 0.10 | 1000 | 0.05 | 4.000 | 0.393722 |
| Aristo-v4 | ✔ | ✔ | 1500 | 0.10 | 1000 | 0.01 | NA | 0.168700 |
| | ✗ | ✔ | 1500 | 0.01 | 500 | 0.01 | NA | 0.307076 |
| | ✔ | ✔ | 1500 | 0.10 | 100 | 0.05 | 0.125 | 0.314443 |

Table 11: Best Hyper-parameter Configurations and the Corresponding Validation MRR for ComplEx Across Datasets with Weighted Relation Perturbation. $d$ stands for embedding size. $lr$ is the learning rate. $bsz$ is the batch size. $reg$ is the regularization strength. $\lambda$ is the weighting of relation prediction. NA indicates not applicable.

## B.2  Hyper-parameter Ranges of Relation Prediction Across Models

We experiment with each model on FB15k-237. Note that the original TucKER [Balazevic et al., 2019] includes some training strategies which are not used in CP, ComplEx and TuckER, like dropout, learning rate decay etc. However, for a fair comparison of how relation prediction affects each model, we trained all the models conditioned on similar settings with Adagrad optimizer and N3 regularisation for at most 400 epochs. We did grid searches and selected the best hyper-parameter configurations according to validation MRR. **We set the weight of relation prediction to 1 in this experiment**. Table 12 lists the grid of the shared hyper-parameters. For RESCAL, the regularisation over predicate matrices can be normalised over the rank to achieve better results. Also F2 regularisation empirically performed better than N3 regulariser for RESCAL. For TuckER, the ranks for predicate and entity are different. Table 13 lists the best hyper-parameter configuration found by our search.

| Model | $d$ or $(d, d_r)$ | $lr$ | $bsz$ | $reg$ |
|---|---|---|---|---|
| RESCAL | [128, 256, 512] | [0.1, 0.01] | [100, 500, 1000] | [0, 0.001, 0.005, 0.01, 0.05, 0.1, 0.5, 1] |
| ComplEx | [100, 500, 1000] | [0.1, 0.01] | [100, 500, 1000] | [0, 0.0005, 0.005, 0.01, 0.05, 0.1, 0.5, 1] |
| CP | [64, 128, 256, 512, 4000] | [0.1, 0.01] | [100, 500, 1000] | [0.005, 0.01, 0.05, 0.1, 0.5, 1] |
| TuckER | [(1000, 150), (1000, 100), (400, 400), (500, 75), (300, 300), (200, 200)] | [0.1, 0.01] | [100, 500, 1000] | [0.005, 0.01, 0.05, 0.1, 0.5, 1] |

Table 12:  Hyper-parameter Search Different KBC Models on FB15k-237. $d$ stands for embedding size. $d_r$ stands for a separate embedding size of relations. $lr$ is the learning rate. $bsz$ is the batch size. $reg$ is the regularization strength.

| Model | Relation Prediction | $d$ or $(d, d_r)$ | $lr$ | $bsz$ | $reg$ | Dev MRR |
|---|---|---|---|---|---|---|
| RESCAL | ✗ | 512 | 0.1 | 500 | 0.00 | 0.365384 |
| | ✔ | 512 | 0.1 | 100 | 0.00 | 0.366789 |
| ComplEx | ✗ | 1000 | 0.1 | 100 | 0.05 | 0.372305 |
| | ✔ | 1000 | 0.1 | 1000 | 0.05 | 0.387133 |
| CP | ✗ | 4000 | 0.1 | 100 | 0.05 | 0.364245 |
| | ✔ | 4000 | 0.1 | 1000 | 0.05 | 0.372408 |
| TuckER | ✗ | (1000, 100) | 0.1 | 100 | 0.10 | 0.358857 |
| | ✔ | (1000, 100) | 0.1 | 100 | 0.50 | 0.359932 |

Table 13:  Best Hyper-parameter Configuration and the Corresponding Validation MRR on FB15k-237 Across Models. For simplicity, we set the weighting $\lambda$ to 1. $d$ stands for embedding size. $d_r$ stands for a separate embedding size of relations. $lr$ is the learning rate. $bsz$ is the batch size. $reg$ is the regularization strength.

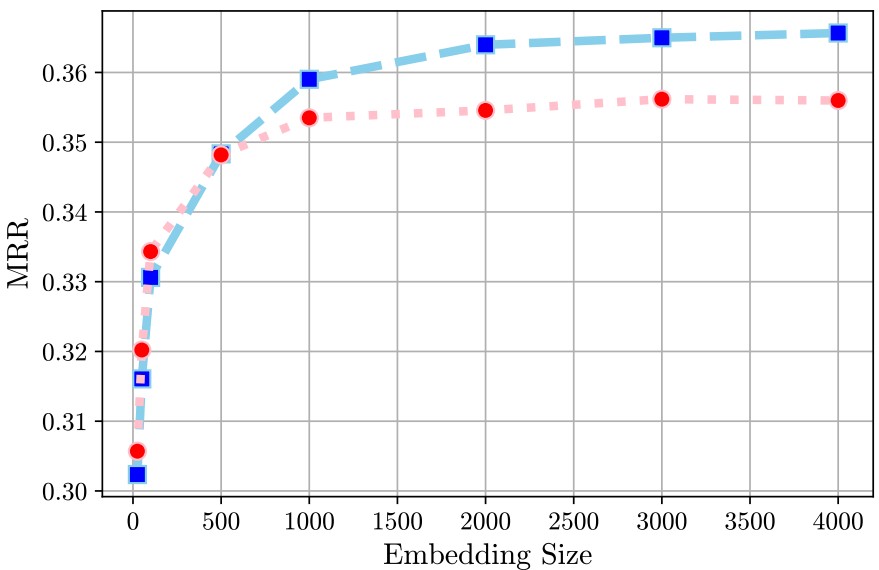

Figure 5: MRR versus Rank for CP on FB15k-237.

## Appendix C. Additional Results

### C.1 More Metrics for Ablation on Rank

Figure 5 (MRR), Figure 6 (Hits@3) and Figure 7 (Hits@10) show the additional metric for the experiments ablating ranks. Blue indicates training with relation prediction while red indicates training without prediction. The range of the rank is $[25, 50, 100, 500, 1000, 2000, 3000, 4000]$

### C.2 Top 20 Queries That Are Improved Most by Relation Prediction

Table 14 shows the top 20 queries of $(?, p, o)$ form that are improved most by relation prediction while Table 15 shows the top 20 queries of $(s, p, ?)$ form that are improved most by relation prediction.

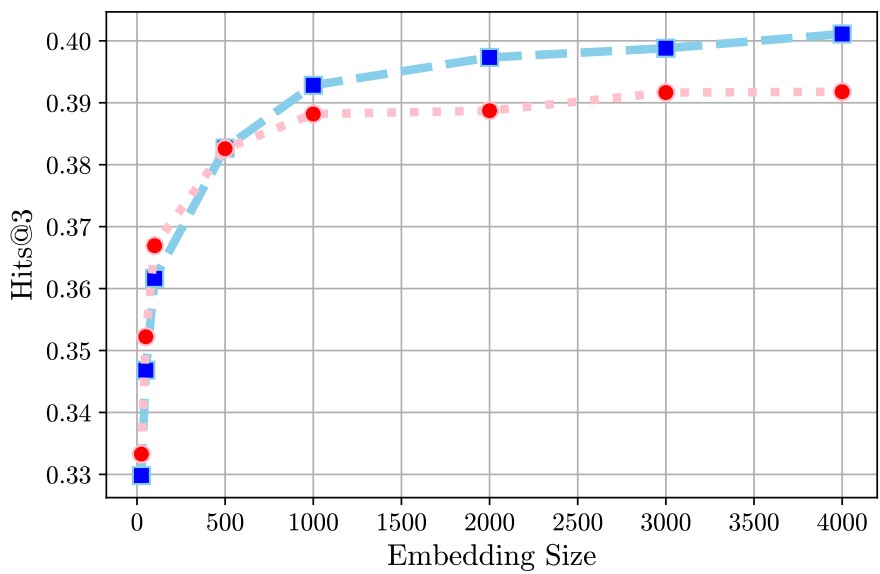

Figure 6: Hits@3 versus Rank for CP on FB15k-237.

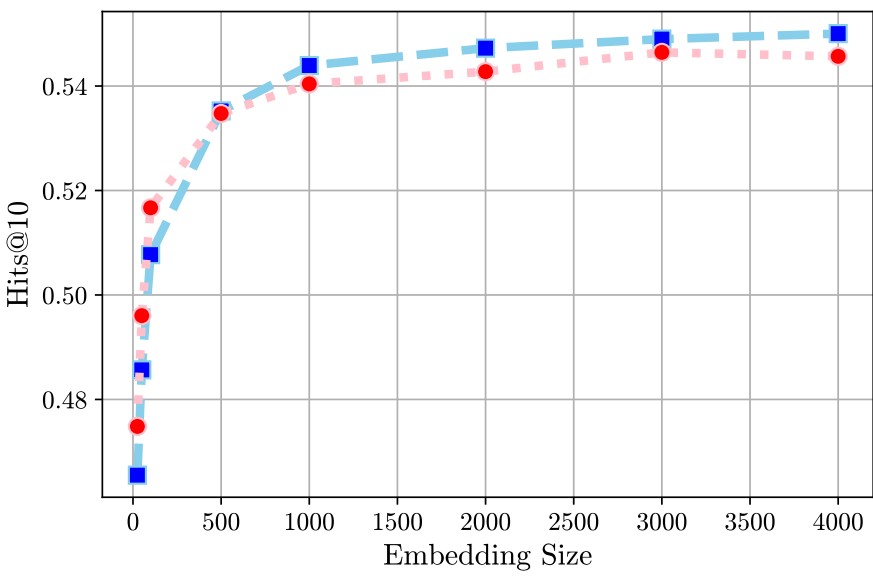

Figure 7: Hits@10 versus Rank for CP on FB15k-237.

| Subject | Predicate | Object | $\Delta$ 1/**Rank** |
|---|---|---|---|
| Paramount Pictures | /common/topic/webpage /common/webpage/category | NA | 1.000 |
| Midfielder | /sports/sports_position/players /sports/sports_team_roster/team | Gaziantepspor | 0.988 |
| The Dictator (2012 film) | /film/film/personal_appearances /film/personal_film_appearance/person | Hillary Clinton | 0.970 |
| Academy Award for Best Supporting Actress | /award/award_category/winners /award/award_honor/award_winner | Maureen Stapleton | 0.963 |
| Christopher Columbus | /user/tsegaran/random/taxonomy_subject/entry /user/tsegaran/random/taxonomy_entry/taxonomy | Library of Congress Classification | 0.950 |
| United Kingdom | /location/location/contains | Westminster | 0.933 |
| President | /organization/role/leaders /organization/leadership/organization | West Virginia University | 0.923 |
| Academy Award for Best Supporting Actor | /award/award_category/winners /award/award_honor/award_winner | Christopher Walken | 0.923 |
| President | /organization/role/leaders /organization/leadership/organization | Bryn Mawr College | 0.917 |
| President | /organization/role/leaders /organization/leadership/organization | Dickinson College | 0.917 |
| National Society of Film Critics Award for Best Actress | /award/award_category/winners /award/award_honor/award_winner | Reese Witherspoon | 0.917 |
| President | /organization/role/leaders /organization/leadership/organization | Louisiana State University | 0.917 |
| Vera Drake | /award/award_winning_work/awards_won /award/award_honor/award | Los Angeles Film Critics Association Award for Best Actress | 0.900 |
| President | /organization/role/leaders /organization/leadership/organization | University of Oklahoma | 0.900 |
| United States | /location/country/second_level_divisions | Marion County, Indiana | 0.900 |
| President | /organization/role/leaders /organization/leadership/organization | University of Southern California | 0.900 |
| Travis Tritt | /film/actor/film /film/performance/film | Blues Brothers 2000 | 0.900 |
| London Film Critics' Circle Award for Director of the Year | /award/award_category/winners /award/award_honor/award_winner | Neil Jordan | 0.900 |
| United States | /location/country/second_level_divisions | Niagara County, New York | 0.900 |
| Deion Sanders | /people/person/places_lived /people/place_lived/location | Atlanta | 0.900 |

Table 14: Top 20 $(s, p, o)$ test triples, based on their increase of the right-hand-side (i.e. on the task of predicting $o$ given $p$ and $s$) reciprocal rank after we introduce the relation prediction auxiliary objective.

| Subject | Predicate | Object | $\Delta$ 1/**Rank** |
|---|---|---|---|
| Midfielder | /soccer/football_team/current_roster /soccer/football_roster_position/position | Wimbledon F.C. | 0.990 |
| Forward (association football) | /soccer/football_team/current_roster /sports/sports_team_roster/position | Iraq national football team | 0.989 |
| United States | /film/film/release_date_s /film/film_regional_release_date/film_release_region | Cleopatra (1963 film) | 0.975 |
| Critics' Choice Movie Award for Best Acting Ensemble | /award/award_nominee/award_nominations /award/award_nomination/award | Matt Damon | 0.975 |
| Female | /people/person/gender | Grey DeLisle | 0.968 |
| United Kingdom | /film/film/release_date_s /film/film_regional_release_date/film_release_region | New Year's Eve (2011 film) | 0.967 |
| United States dollar | /location/statistical_region/rent50_2 /measurement_unit/dated_money_value/currency | Anchorage, Alaska | 0.966 |
| United Kingdom | /film/film/release_date_s /film/film_regional_release_date/film_release_region | Killing Them Softly | 0.962 |
| Glendale, California | /people/person/spouse_s /people/marriage/location_of_ceremony | Jane Wyman | 0.962 |
| United Kingdom | /film/film/release_date_s /film/film_regional_release_date/film_release_region | ParaNorman | 0.962 |
| Streaming media | /film/film/distributors /film/film_film_distributor_relationship/film_distribution_medium | Pulp Fiction | 0.960 |
| United Kingdom | /film/film/release_date_s /film/film_regional_release_date/film_release_region | Magic Mike | 0.958 |
| United States dollar | /location/statistical_region/rent50_2 /measurement_unit/dated_money_value/currency | Napa County, California | 0.952 |
| United Kingdom | /film/film/release_date_s /film/film_regional_release_date/film_release_region | Rock of Ages (2012 film) | 0.952 |
| United Kingdom | /film/film/release_date_s /film/film_regional_release_date/film_release_region | Contact (1997 American film) | 0.950 |
| Streaming media | /film/film/distributors /film/film_film_distributor_relationship/film_distribution_medium | American History X | 0.950 |
| Los Angeles | /organization/organization/place_founded | Paramount Pictures | 0.947 |
| United Kingdom | /film/film/release_date_s /film/film_regional_release_date/film_release_region | L.A. Confidential (film) | 0.941 |
| Psychological thriller | /film/film/genre | Family Plot | 0.941 |
| United Kingdom | /film/film/release_date_s /film/film_regional_release_date/film_release_region | Moneyball (film) | 0.938 |

Table 15: Top 20 $(s, p, o)$ test triples, based on their increase of the left-hand-side (i.e. on the task of predicting $s$ given $p$ and $o$) reciprocal rank after we introduce the relation prediction auxiliary objective.