# OpenReview forum: "Relation Prediction as an Auxiliary Training Objective for Improving Multi-Relational Graph Representations"
_AKBC.ws/2021/Conference — AKBC 2021_

### Official Review · Reviewer_UiqN · 2021-07-12
**A thorough empirical analysis of the effect of an overlooked training objective (relation perturbation), surprising results**

**Rating:** 8
**Confidence:** 5

**Review:**

The paper tries to answer the following question - does knowledge graph link prediction improve when relation prediction is added as an additional training task?

The authors add the relation prediction objective (along with subject/object prediction objective) while training multiple SOTA link prediction models. They perform an extensive hyperparameter search over 6 datasets (5 of which are standard datasets), resulting in over 40,000 experimental runs. The results show that for most datasets, a new state-of-the-art for link prediction is established by models that incorporate relation prediction during training.

Pros:
1. In my knowledge, relation prediction during training has been used in the KG embedding space only when evaluating on the relation prediction task. This might be the first instance of 'multi-task training' in this space.
2. Extensive hyperparameter search has been performed, which is essential for proper comparison of models [Ruffinelli et al ICLR 2020]. Statistical significance has also been reported.

Cons:
1. It might be interesting to see performance on link prediction while training *only* with relation prediction loss. Although I don't expect this to perform better than the full loss currently used, it is an ablation to see whether using all losses together is essential or relation prediction loss is the sole reason for SOTA performance.

Overall, the paper is well written. The results are promising and give a further direction to KG Embedding research ie. using additional self-supervised training objectives.

---

> ### Author Response · Authors · 2021-07-30
> **Added experiments for training only with relation prediction**
>
> Thank you for your review. We run the suggested experiments training with relation-only and update our results in the revised version. The results are available in Tab. 2 and Tab. 3. It is interesting that relation-only self-supervised training leads to better results on Kinship compared to training with the commonly used entity prediction task. As expected, on the remaining datasets, the link prediction accuracy when training only with a relation prediction objective is significantly worse than training with both relation prediction and entity prediction objectives.

---

### Official Review · Reviewer_Ks9P · 2021-07-16
**Nice paper with small contribution**

**Rating:** 7
**Confidence:** 3

**Review:**

The paper proposes to extend the objective function for KG-embedding learning with a term that predicts the relation (predicate) given two entities. This extension is well motivated and the experiments show that it leads to small but consistent improvements. While the technical contribution of the paper is rather small, the experimental setup is sound and there are many interesting results and analyses. Especially the clear statement and answer of the research questions is nice.
One aspect that should be elaborated more on is the choice of lambda: It seems that for some experiments and analyses, lambda was tuned and for others it was not. This should be made consistent and explained carefully as it could have a large impact on the results.

Other comments:
- It is confusing that the authors use "rank" for both the rank of entities in the experimental results and for the embedding dimensionality. I would recommend to use "dimensionality" for the latter as (i) this is standard terminology, (ii) the dimensionality of the embedding matrix does not necessarily correspond to its rank.
- The connection to masked-language-model pretraining seems to be quite artificial especially since MLM does not mask out every token but only a random subset of them, etc.
- Section 2 should include a better connection to the rest of the paper: How did the authors select the methods they applied? Why didn't the authors describe TransE nor use it as another baseline in their experiments?
- The second paragraph of "Training Objectives" (page 3) should be rewritten: First, 2018 is not later than 2019/2020. Second, KvsAll and 1vsAll is described using the same sentence.
- Section 4.3: I don't see why a t-SNE plot should help with analyzing many-to-many setups. This needs a better motivation.
- There are some writing inconsistencies, e.g., "hyperparameter" vs. "hyper-parameter"
- "T-SNE" is typically written as "t-SNE"

Question to the authors:
- Section 4.1: Do you have an intuition why your model is worse for Hits@3 and Hits@10 on UMLS?

---

> ### Author Response · Authors · 2021-07-30
> **Changed terminology of "rank" to "embedding size", revised section 2, explanation on UMLS**
>
> Thank you for your insightful comments and questions!
> Here are our answers:
>
> 1. We agree with you that the terminology of "rank" is indeed inappropriate: we solved this in the revised version by referring to the “embedding size” instead.
>
> 2. You are right: we toned down the connection to MLMs in the latest revision.
>
> 3. Thank you for pointing this out: we revised Section 2 to better connect it with the rest of the paper. About our model choices, we noted in Section 4 that we focus on RESCAL, ComplEx, CP, and TuckER since they have been proven to be very competitive and reproducible (Ruffinelli et al., 2020). We clarify this aspect in the revised version of the paper.
>
> 4. Thank you for noticing this -- indeed the sentences describing 1vsAll and KvsAll are quite similar except for a single word (multi-class, multi-label): we clarified and expanded our description of such two training objectives, and revised the order of the references.
>
> 5. We used t-SNE visualisations since we wanted to analyse whether introducing relation prediction helped to learn more diverse predicate representations and the clusters in the t-SNE plot serves as a proxy for telling the difference between predicates.  We elaborated on the clusters presented in the visualisations by showing samples of predicates from each cluster in Table 7. In addition, in Table 6, we show the top 10 predicates whose link prediction results benefited the most by the introduction of the relation prediction objective.
>
> 6. Thanks, we addressed the typos.
>
> 7. Thank you for noting the issue about the tuning of the lambda hyper-parameter: we updated the paper to explain this carefully. For the main results on all the datasets, we tuned lambda using grid-search.  For the ablation experiments on the number of predicates and for different choices of models, we did not tune lambda due to computational constraints on our side. Setting lambda to 1, however, already yields significant improvements. Furthermore, tuning it yields more accurate link prediction results, at the cost of having to tune one more hyper-parameter. Details about the hyper-parameter grid can be found in Appendix B.
>
> 8. In our experiments, we observe that generally relation prediction improves Hits@1 more than Hits@3/10, not only on UMLS datasets. On the reasons why relation prediction seems to yield better results in terms of MRR and Hits@1, but not in terms of Hits@3 and Hits@10: we believe this happens because, for multiple queries, the correct answer entities are already among the top-k results (where k is 3 or 10), and introducing relation prediction allows them to be in the top-1, thus improving Hits@1 and MRR.

---

### Official Review · Reviewer_wSBo · 2021-07-22
**Simple idea that works well**

**Rating:** 7
**Confidence:** 4

**Review:**

The paper introduces a new self-supervised objective for improving link prediction in knowledge graphs. The idea presented in the paper naturally extends previous self-supervised objectives (e.g., TransE, CP, DistMult, etc.) by adding a training objective to predict the relation given the head and the tail entity in a knowledge graph triplet. The idea is really simple and it is surprising that previous works haven't tried this loss in addition to predicting the head/tail entity given the relation and the other entity.

The paper performs a comprehensive set of experiments that show that the proposed objective improves link prediction (based on MRR, Hits@k) on various standard benchmark datasets. Analysis shows that the proposed technique provides larger improvements on KGs that contain more relations, which is expected and good to corroborate experimentally.

Overall, I like the simplicity of the idea presented in the paper and the comprehensive nature of the experiments presented that show that the relation prediction objective helps in link prediction.

I do not find any concerning weaknesses with the paper but I do find some parts of the writing/detail concerning or unclear. I will suggest the authors to please address these in the next iteration of the paper:

1. In Table 2, I do understand that the performances reported can be from different models, but is at least any full row based on the same model or each cell in the table can be from a different one? For example, Kinship (X) contains numbers 0.916, 0.866, 0.964, and 0.988. Are all these performances from the same model or different? If different, I do find this table problematic. If the same model, the authors should specify which model it is and based on which metric was that model chosen.

2. The intuition connecting the proposed technique with the MLM objective in BERT seems too far-fetched. A knowledge graph triplet has enough differences with a sequence of text tokens that the intended connection does not make sense. I would suggest removing mentions of this.

3. I did not understand the motivation behind and the outcome of the analysis performed using the tSNE plots. I think that drawing conclusions based on tSNE plots is not a good idea; tSNE transformation to low-embedding spaces is enough lossy that makes any conclusions drawn from it meaningless.

4. Sec 4.2.1 - I think the authors mean the embedding size of the entities/relations and not the rank of the embedding matrix. Though the rank is upper-bounded by this size, it could very well be much lower.

5. (minor) In Table 1, the authors use dot (.) to denote digit grouping while I think it is standard practice in most scientific literature to use commas (,) or just spaces (induced by \, in latex).

---

> ### Author Response · Authors · 2021-07-30
> **Clarification for hyper-parameters and t-SNE**
>
> Thank you for your valuable feedback and comments!
> Regarding your questions:
>
> 1. Yes, results in each row of Table 2 were computed with the same model: we clarified this both in the table and in its description. The hyper-parameter search process is described in Section B.1 (Table 7) in the Appendix.
>
> 2. Thanks for pointing this out: we toned down the reference to MLMs in the updated version of the paper. Our main point is that including relation prediction relaxes the restrictions on the positions of the masked symbol in the triple, by not limiting it to be either the subject or the object.
>
> 3. We agree with you regarding t-SNE in that transformations to 2D spaces can be lossy. We used t-SNE visualisations since we wanted to analyse whether introducing relation prediction helped to learn more diverse predicate representations and the clusters in the t-SNE plot serves as a proxy for telling the difference between predicates. We elaborated on the clusters presented in the visualisations by showing samples of predicates from each cluster in Table 7. In addition, in Table 6 we show the top 10 predicates whose link prediction results benefited the most by the introduction of the relation prediction objective.
>
> 4. Thank you: we replaced “rank” with “embedding size” in the revised version.
>
> 5. Thank you for pointing this out: we solved this in the revised version.

---

> > ### Comment · Reviewer_wSBo · 2021-07-31
> > **Thanks! t-SNE still not useful**
> >
> > Thank you for your response.
> >
> > The t-SNE plot is still not really helpful, I think. In Table 7, you classified `nationality` and `religion` as hierarchy but not `student` or `major field of study`. I could argue that these relations are similar to nationality or religion in more than ways that one. I think that conclusions drawn from t-SNE should be taken with a pinch of salt.
> >
> > Is there any intuition why the performance improves on the relations in Table 6? Does it have to do something with the in-/out-degree of the node that participates in these relations? Especially 3 of top 10 are position relations in baseball, football and hockey. Could you please give some examples of (subject, object) entities for those relations?
> >
> > My rating for the paper remains the same.

---

### Author Response · Authors · 2021-07-30
**Thank you!**

Thank you for your insightful comments and questions!
We uploaded a revised version of our paper and answered each reviewer.

---

### Decision · Program_Chairs · 2021-08-18

**Decision:**

Accept

**Comment:**

This paper proposes a self-supervised extension to the KG embedding objective function to predict relation predicate given two entities. The extension is simple, well motivated and results in small but consistent improvements across datasets. Extensive hyper-param search validates their improvements with statistical significance and sets SOTA results on multiple link prediction datasets. Reviewer comments are well answered in author response. There is a small concern about "t-SNE visualizations being not so helpful in analyzing many-to-many setups", this part can be clarified further in the camera ready. No other major concerns.